Population genetics of swamp eel in the Yangtze River: comparative analyses between mitochondrial and microsatellite data provide novel insights

Zhou Huaxing
Hu Yuting
Jiang He kenc7c7c7@126.com
Duan Guoqing
Ling Jun
Pan Tingshuang
Chen Xiaolei
Wang Huan
Zhang Ye
Anhui Key Laboratory of Aquaculture and Stock Enhancement, Fisheries Research Institution, Anhui Academy of Agricultural Sciences , Hefei , China
Rollins Lee
Electronic publication date: 2020 Jan 21
Publication date: 2020
Volume: 8
Electronic Location ID: e8415
Received 2019 Jul 31; Accepted 2019 Dec 16
Copyright: ©2020 Zhou et al.
Copyright year: 2020
Copyright holder: Zhou et al.
License: This is an open access article distributed under the terms of the Creative Commons Attribution License, which permits unrestricted use, distribution, reproduction and adaptation in any medium and for any purpose provided that it is properly attributed. For attribution, the original author(s), title, publication source (PeerJ) and either DOI or URL of the article must be cited.
License URL: https://creativecommons.org/licenses/by/4.0/

Keywords: Monopterus albus, Sex reversal, Mitochondrial and nuclear markers, Population genetics

Funding: Earmarked Fund of Anhui Fishery Research System No. 2016-84 Anhui Academy of Agricultural Sciences Technology Innovation Team No. 2019YL026 Chinese agriculture research system No. CARS-48 This work was funded by the Earmarked Fund of Anhui Fishery Research System (No. 2016-84), Anhui Academy of Agricultural Sciences Technology Innovation Team (No. 2019YL026) and Chinese agriculture research system (No. CARS-48). The funders had no role in study design, data collection and analysis, decision to publish, or preparation of the manuscript.

==============================
The swamp eel (Monopterus albus) is a typical sex reversal fish with high economic value. Several phylogeographic studies have been performed using various markers but comparative research between mitochondrial and nuclear markers is rare. Here, a fine-scale study was performed across six sites along the Yangtze River including three sites on the main stem and three sites from tributaries. A total of 180 swamp eel individuals were collected. Genetic structure and demographic history were explored using data from two mitochondrial genes and eight microsatellite loci. The results revealed the samples from tributary sites formed three separate clades which contained site-specific lineages. Geographic isolation and the habitat patchiness caused by seasonal cutoff were inferred to be the reasons for this differentiation. Strong gene flow was detected among the sites along the main stem. Rapid flow of the river main stem may provide the dynamic for the migration of swamp eel. Interestingly, the comparative analyses between the two marker types was discordant. Mitochondrial results suggested samples from three tributary sites were highly differentiated. However, microsatellite analyses indicated the tributary samples were moderately differentiated. We conclude this discordance is mainly caused by the unique life history of sex reversal fish. Our study provides novel insights regarding the population genetics of sex reversal fish.

Introduction

The swamp eel (Monopterus albus) is a typical sex reversal fish, belonging to the family Synbranchidae, which usually inhabits swamps, ponds and rice fields (Nelson, Grande & Wilson, 2016). Due to its high nutritional value and good taste, the swamp eel is used as a significant aquatic food in China. In 2018, 0.32 million tons of swamp eel were produced by Chinese aquaculture (data from The Ministry of Agriculture of China, 2019).

With the development of swamp eel farming, population genetic research of the wild eel became a hot topic, which has been used to guide the genetic breeding and swamp eel aquaculture. Several studies have used different markers including microsatellites, mitochondrial sequences and Inter-Simple Sequence Repeats (ISSR) to explore swamp eel population genetics (Lei et al., 2012; Li et al., 2013; Liang et al., 2016). Previous studies suggested a rapid decrease of wild swamp eel caused by the large use of pesticides and over-exploitation (Li et al., 2013; Liang et al., 2016). Cultured populations were genetically less diverse than wild populations. Due to farm escapes, the populations of wild swamp eel suffer from genetic homogenization and degeneration of genetic characters (Li et al., 2013). Thus, further research of population genetics and dynamics are necessary to protect the wild swamp eel.

Monopterus albus is a sex reversal fish with a unique life history (Liu, 1944; Mazzoni et al., 2018; Qu, 2018). It starts the reproductive cycle as a functional female. During 1 to 1.5 years of age, females obtain well-developed ova. After spawning, the swamp eel’s sex is reversed from female to male. The intersexes appear in the two-year-old age class. Then it lives as male without sex reversal (Liem, 1963). Generally, swamp eels can live to 6-8 years. Mitochondrial DNA is maternally inherited. For this reason, the inheritance patterns of protogynous hermaphrodites differ to that of species that do not practice sex reversal (Coscia et al., 2016). Therefore, it may be important to include both nuclear and mitochondrial markers to explore the population genetics of sex reversal fish.

This fine-scale study was performed across six sites along the Yangtze River including three sites on the main stem and three sites from tributaries. Subsequently, the genetic structure of these six sampling sites was explored, using sequence from two mitochondrial genes and eight microsatellite loci. Here, we show how different types of molecular markers can provide new insights regarding the population genetics of sex reversal fish.

Materials & Methods

Ethics statement and Sample collection

Procedures involving animals and their care were approved by the Animal Care and Use Committee of Anhui Academy of Agricultural Sciences under approval number 201003076. Field experiments were approved by Fisheries Bureau of Anhui (project number: FB/AH 2017-10).

A total of 180 individuals were collected using net from six sites along the Anhui basin of the Yangtze River (Table S1). Sampling sites from tributaries of the Yangtze River included Dang Tu (DT), Fan Chang (FC) and Huai Ning (HN); sampling sites from the main stem of the Yangtze River included Wu Wei (WW), Gui Chi (GC) and Wang Jiang (WJ) (Fig. 1).

Figure 1 Sampling sites along the Yangtze River mapped using DIVA-GIS.

Three sampling sites from main stem included Wu Wei (WW), Gui Chi (GC) and Wang Jiang (WJ); three sampling sites from tributaries included Dang Tu (DT), Fan Chang (FC) and Huai Ning (HN). FC tributary connected to the main stem during the wet season and isolated during the dry season. Data source: DIVA-GIS (http://swww.diva-gis.org/Data).

DNA extraction and Marker genotyping

Total genomic DNA was extracted from muscle tissue using a standard phenol/chloroform procedure via proteinase K digestion (Sambrook, Fritsch & Maniatis, 1989), and then kept at −20 °C for PCR amplification.

The mitochondrial cytochrome c oxidase subunit I (COI) gene and cytochrome b (Cyt b) gene were chosen. Two pairs of primers were designed here for the amplification (Table S2). PCR were conducted in 50 µL reaction mixtures containing 200 ng template DNA, 5 µL 10 × buffer (TaKaRa, Dalian, China), 4 µL MgCl2 (2.5 mol/L), 3 µL dNTP (2.5 m mol/L), 2 µL of each primer (5 µmol/L), and 1 U Taq DNA polymerase (5 U/ µL, TaKaRa). PCR conditions were as follows: initial denaturation (95 °C, 1 min), then 35 cycles of denaturation (94 °C, 50 s), primer annealing (55 °C, 45 s), and elongation (72 °C, 1 min) and a final extension (72 °C, 10 min). All fragments were sequenced in both directions with an ABI3730 automated sequencer (Invitrogen Biotechnology Co., Ltd, USA). Then these two gene sequences were combined for subsequent analyses.

Eight unlinked polymorphic microsatellite loci were selected from previous studies (Table S1) (Lei et al., 2012; Li et al., 2007; Zhuo et al., 2011). PCR were conducted in 50 µL reaction mixtures containing 200 ng template DNA, 5 µL 10 × buffer (TaKaRa, Dalian, China), 4 µL MgCl2 (2.5 mol/L), 2.5 µL dNTP (2.5 m mol/L), 2 µL of each primer (5 µmol/L), and 1 U Taq DNA polymerase (25 U/ µL, TaKaRa). PCR conditions were as follows: initial denaturation (95 °C, 5 min), then 32 cycles of denaturation (94 °C, 30 s), primer annealing (57 °C, 60 s), and elongation (72 °C, 90 s) and a final extension (72 °C, 5 min). Genotypes were detected by ABIPRISM 3730.

Data analyses

Mitochondrial sequence

Sequences were assembled by DNASTAR Lasergene package. Subsequent homologous alignment was performed by Mafft v.7 online program (https://mafft.cbrc.jp/alignment/software/) (Katoh, Rozewicki & Yamada, 2019).

Haplotype and nucleotide diversity were estimated using DNAsp V.6 (Rozas et al., 2017). A haplotype network was constructed using Median Joining (MJ) in NETWORK v.5.0 (Bandelt, Forster & Röhl, 1999). Analysis of Molecular Variance (AMOVA) was performed using Arlequin v.3.11 (Excoffier, Laval & Schneider, 2005). Genetic variation within and among sampling sites was assessed. Pairwise Fst was estimated in order to evaluate the levels of population differentiation (Slatkin & Barton, 1989) and the P values were corrected using multiple testing.

The demographic history was explored using three approaches, e.g., neutrality tests, mismatch distribution and Bayesian Skyline Plots (BSP) analyses. Tajima’s D (Tajima, 1989) and Fu’s Fs (Fu, 1997) values were calculated using DNAsp V.6.Mismatch distribution analyses were performed using Arlequin v.3.11 (Rogers & Harpending, 1992). The expansion time was calculated by the τ value with the equation τ=2 µt, where µrepresents the nucleotide mutation rate and t represents the estimated expansion time. BSP analysis was performed using Beast v1.10.4 (Suchard et al., 2018) under an uncorrelated relaxed clock mode for 5 × 107 generations.

Microsatellites data

The results of 8 microsatellites loci were read using GeneMarker (Holland & Parson, 2011) and reformatted using Convert v.1.31 (Glaubitz, 2004). Hardy-Weinberg equilibrium (HWE) tests were performed using Popgene v 1.32 (Yeh et al., 1997). Expected and observed heterozygosity were calculated with Arlequin v.3.11 (Rogers & Harpending, 1992). AMOVA was implemented with Arlequin. Pairwise Fst was computed based on Slatkin’s method (Slatkin & Barton, 1989). The geographical and genetic distance between sample sites was measured by GPS and Popgene v 1.32, respectively. The correlation between geographical and genetic distance was analyzed using Pcord v 5 (Grandin, 2006).

Population structure was estimated using an MCMC (Markov Chain Monte Carlo) algorithm as implemented in Structure v.2.3.3 (Hubisz et al., 2009). The number of clusters (K) was calculated under 1 × 106 iterations with 10 replications and the optimal number of K was deduced by Structure Harvester Web v.0.6.94 (Evanno, Regnaut & Goudet, 2005; Earl, 2012).

Results

Mitochondrial genes

A total of 1752 bp of mitochondrial sequence (COI 665 bp, Accession number: MN097948–MN098127; Cyt b 1087 bp, Accession number: MN098128–MN098307) were obtained for analyses. The contents of the bases A, T, G and C were 24.6%, 29.3%, 14.6% and 31.5% respectively, which showed obvious anti-G bias (Saccone et al., 1999).

The 180 mitochondrial sequences corresponded to 86 distinct haplotypes (Table 1). All haplotypes were divided into four clades based on MJ method (Fig. 2). Clade A was the largest one which contained samples from five sampling sites. Haplotype 5 (H-5) had the largest number of shared individuals, and its central placement in the network suggests that this is the ancestral haplotype. The other three clades were separated by 13, 47, 24 mutational steps, respectively. Clade B and C only contained samples from the HN and FC sampling sites, respectively. Clade D mainly consisted of DT samples.

Table 1 Genetic diversity of samples from six sites assessed using mitochondrial and microsatellite data.

Sampling sites	Mitochondrial genes	Microsatellite loci	
	Num. of haplotypes	Hd	Pi	Num. of alleles	He	Ho	
DT (Tributary)	18	0.9540	0.0148	14	0.8749	0.7917	
WW (Main stem)	24	0.9793	0.0078	14	0.8638	0.7292	
FC (Tributary)	9	0.6620	0.0017	12	0.8089	0.6542	
GC (Main stem)	15	0.8480	0.0021	13	0.8420	0.7417	
HN (Tributary)	13	0.9220	0.0072	12	0.8052	0.5958	
WJ (Main stem)	19	0.9240	0.0023	14	0.8516	0.7802	
Notes. Hd represents haplotype diversity; Pi represents nucleotide diversity; He represents expected heterozygosity; Ho represents observed heterozygosity.

Figure 2 Haplotype network showing the genetic relationship of samples using Median Joining (MJ) method.

Different colors represent the six populations. Circle size represents the number of sequence. The largest circle represents n = 24 and the smallest circle represent n = 1. Numbers of nearby branches represent the mutational steps and no numbers represent only one mutational step. Black dots represent Median Vector (mv).

Haplotype diversity ranged from 0.6620 to 0.9793 and nucleotide diversity ranged from 0.0017 to 0.0148 based on mitochondrial sequence (Table 1). The results of AMOVA showed that genetic variation among sampling sites (71.23%, P < 0.001) were much higher than the variation within the sampling sites (28.77%, P < 0.001) (Table 2A). Subsequent Fst values further confirmed this result. Strong gene flow was detected between the main stem sampling sites (Fst = 0.0242 between GC and WW, Fst = 0.0286 between WJ and WW, Fst = 0.0305 between WJ and GC, see Table 3A). And high differentiation was revealed between the tributary sampling sites (Fst = 0.3069 − 0.9431) (Table 3A). Fu’s Fs and Tajima’s D tests of main stem samples were significant (P < 0.01) but negative. No explicit expansion or decline were revealed for the tributary samples and the Fu’s Fs and Tajima’s D values except the Tajima’s D of FC were not significant (Table 4). Mismatch distribution analysis revealed similar results. The values of sum of squares deviations (SSD) for samples from main stem and DT were not significant (P > 0.05), indicating that sudden expansion could not be rejected (Table 4). The BSP analysis suggested the main stem samples had expanded roughly in 0.46 MYA (Fig. 3).

Table 2 AMOVA of 6 sampling sites indicating the source of variation.

Table 2a AMOVA of 6 sampling sites indicating the source of variation by mitochondrial data used.	
Source of variation	df	Sum of squares	Percentage of variation	P value	
Among sampling sites	5	1965.433	71.23	<0.001	
Within sampling sites	174	908.533	28.77	<0.001	
Total	179	2873.967			
Table 2b AMOVA of 6 sampling sites indicating the source of variation by microsatellite loci used.	
Source of variation	df	Sum of squares	Percentage of variation	P value	
Among sampling sites	5	76.325	5.35	<0.001	
Within sampling sites	352	672.105	94.65	<0.001	
total	357	1260.43			

Table 3 Pairwise values of Fst (below diagonal) and P (above diagonal) between sampling sites.

Table 3a Pairwise values of Fst (below diagonal) and P (above diagonal) between sampling sites estimated using mitochondrial data.	
	DT	WW	FC	GC	HN	WJ	
DT		<0.01	<0.01	<0.01	<0.01	<0.01	
WW	0.6228		<0.01	0.08	<0.01	0.05	
FC	0.5800	0.8510		<0.01	<0.01	<0.01	
GC	0.7296	0.0242	0.9431		<0.01	<0.01	
HN	0.6441	0.3069	0.8551	0.4750		<0.01	
WJ	0.7253	0.0286	0.9391	0.0305	0.4594		
Table 3b Pairwise values of Fst (below diagonal) and P (above diagonal) between sampling sites estimated using microsatellite loci.	
	DT	WW	FC	GC	HN	WJ	
DT		<0.01	<0.01	<0.01	<0.01	<0.01	
WW	0.0653		<0.01	0.06	<0.01	0.37	
FC	0.0813	0.0509		<0.01	<0.01	<0.01	
GC	0.0794	0.0077	0.0585		<0.01	0.63	
HN	0.0982	0.0597	0.0869	0.0636		<0.01	
WJ	0.0704	0.0030	0.0509	0.0005	0.0493		

Table 4 Summary of neutrality and mismatch analyses indicating the demographic history.

Populations	Tajima’s D	Fu’s Fs	τ	SSD (P)	
Main stem	−2.3328**	−24.9078**	2.2949	0.0056(0.36)	
DT	0.6878	1.8735	0.3906	0.0344(0.53)	
FC	−2.4163**	−0.8094	0.375	0.1446(0.02)	
HN	0.4788	2.1653	22.5293	0.0496(0.008)	
Notes.

** P < 0.01.

Figure 3 The demographic history inferred from mitochondrial data.

Samples from three main stem sampling sites were treat as one group. (A) Mismatch distribution of main stem samples; (B) Bayesian skyline plots of main stem samples, the shaded area represents the 95% confidence intervals of Highest Posterior Density (HPD) analysis; (C) Mismatch distribution of DT samples; (D) Bayesian skyline plots of DT samples; (E) Mismatch distribution of FC samples; (F) Bayesian skyline plots of FC samples; (G) Mismatch distribution of HN samples; (H) Bayesian skyline plots of HN samples.

Microsatellite loci

The eight microsatellite loci amplified unambiguous and repeatable products in the size range expected. All loci were in Hardy-Weinberg equilibrium (P > 0.05). High genetic diversity was also supported by microsatellite data. Expected and observed heterozygosity for the six sampling sites were 0.8052 –0.8749 and 0.5958 –0.7917, respectively (Table 1).

Structure results suggested the highest posterior probability for K = 4 (Fig. S1). The ΔK method revealed four potential genetic clusters, aligning with the three tributaries and all the main stem sites together. Samples from main stem showed high levels of genetic admixture (Fig. 4).

Figure 4 Population structure of 180 swamp eels showing for K = 4.

Four colors, e.g., red, yellow, purple and green, represent the inferred genetic clusters.

AMOVA was performed using the microsatellite data and suggested that genetic variation was mainly within sampling sites (94.65%, P < 0.001), opposite to the results from mitochondrial data (Table 2B). Fst values suggested low levels of differentiation (Fst = 0.0005 − 0.0982) (Table 3B).

The correlation between genetic and geographic matrixes was assessed using Mantel test (Table S3). The results suggested a significant correlation between them (r = 0.8791, P = 0.004) (Fig. 5).

Figure 5 Plotmatrix indicating the correlation between genetic and geographic matrixes.

Discussion

Population genetics of swamp eel

High levels of genetic diversity were found across sampling sites at both mitochondrial and microsatellite markers. The genetic diversity level of this study except FC sample site was higher than previous study in the same basin (Hd = 0.708 and Pi = 0.002 based on mitochondrial D-loop sequences) (Liang et al., 2016). The genetic diversity of FC samples was the lowest (Hd = 0.6620, Pi = 0.0017). The significant differentiation of three tributary sampling sites was revealed by the population genetic analyses (Fst >0.25). The haplotype network and structure results suggested the tributary samples formed three separate clades which contained site-specific lineages. Significant correlation between genetic and geographic distance was detected. Interestingly, strong gene flow was detected among the main stem sampling sites and the expansion of main stem samples was detected.

It is well known that the swamp eel is a burrowing fish whose fins are vestigial or absent (Nelson, Grande & Wilson, 2016). Compared with most fishes, the swimming ability of eel is weak. Thus, we were curious about the reasons for this long-distance gene flow among main stem sampling sites. The flow rate of main stem in Anhui basin range up to 1.0 m/s (Guo & Xia, 2007). Rapid flow provides the dynamic for the migration of swamp eel. The eggs and juvenile fishes can slip downstream to the farther places. However, due to the flat stream gradient and curved channel, the tributary flow becomes slower (Zhang, YT & Jiang, 2008) and long-distance migration is difficult for swamp eel. We propose that geographic isolation and the habitat patchiness caused by seasonal cutoff are the reasons for the differentiation between tributary samples. The tributary of FC site was much more isolated from the main stem. It connected to the main stem during the wet season and isolated during the dry season. Long-term isolation from the main stem may cause the low genetic diversity of FC samples.

Comparative analyses between mitochondrial and microsatellite data

Analyses of nuclear and mitochondrial markers revealed discordant population structure. Based on mitochondrial data, genetic variation was mainly found among sampling sites. Samples between the tributary sites were highly differentiated (Fst > 0.25) and represented three monophyletic clades. However, microsatellite analyses suggest that the majority of genetic variation is within these sampling sites; samples between the tributary sites were moderately differentiated (0.05 < Fst < 0.15). Mean Fst values among six sampling sites based on mitochondrial and microsatellite data were 0.548 and 0.055, respectively. The mean mitochondrial Fst value (0.548) was almost ten times higher than the Fst (0.055) estimated with microsatellite data.

Our study provided an interesting pattern of discordance between markers for population genetics. According to previous studies, sex-biased dispersal, genetic admixture and lineage sorting may be the potential reasons for the discordance caused by different molecular markers (Funk & Omland, 2003; Qu et al., 2012; Yang et al., 2016; Zarza, Reynoso & Emerson, 2011). Considering the sex reversal in this species, we inferred the unique life history of the swamp eel contributed to this discordance. Initially, the swamp eel is female and provides both mitochondrial and nuclear DNA to the population genetic pool. After spawning, the swamp eel becomes male. Male swamp eels are much bigger and stronger than females. Males could migrate farther and have a higher survival rates, which provides a potential of male sex-biased dispersal. Due to mitochondrial maternal inheritance, male swamp eels only provide nuclear DNA to the population genetic pool (Fig. 6). So male sex-biased dispersal may cause the differences in population structure between the markers. As mentioned above, male stage is much longer than female stage in the whole life of swamp eel that may cause different genetic frequencies between mitochondrial and nuclear data. The ten–fold difference between mitochondrial and nuclear Fst values also confirmed this hypothesis.

Figure 6 Diagram showing the unique life history and different hereditary patterns of mitochondrial DNA and nuclear DNA in sex reversal fish.

Conclusions

Our study used two data sets, mitochondrial DNA and microsatellites, to explore the demography, genetic variation and population structure of swamp eels. Compared with previous studies, high levels of genetic diversity suggest that swamp eels are an abundant resource in the Anhui basin and have potential commercial value. Samples from each tributary site in this study should be treated as an independent genetic unit. The unique sex reversal life history of the swamp eel may be a significant factor affecting the population genetic structure and may generate the discordance we found between different molecular markers. Our study provides novel insights regarding the population genetics of sex reversal fish.

Supplemental Information

Figure S1 Δ K values for K ranging from 1 to 10, indicating the optimal K = 4

Click here for additional data file.

Table S1 Sampling information of each site in this study

Click here for additional data file.

Table S2 The primer information of two mitochondrial genes and eight microsatellite loci

Click here for additional data file.

Table S3 The genetic (below diagonal) and geographic (above diagonal) (km) distance matrixes

Click here for additional data file.

Supplemental Information 1 The raw data of microsatellite loci

Click here for additional data file.

Supplemental Information 2 Network raw data

Click here for additional data file.

Supplemental Information 3 The mitochondrial cytochrome c oxidase subunit I (COI) gene (MN097948–MN098127)

Click here for additional data file.

Supplemental Information 4 The mitochondrial cytochrome b (Cyt b) gene (MN098128–MN098307)

Click here for additional data file.

We thank Assoc. Prof. Lee Rollins, Adomas Ragauskas, Joana Robalo and Dr. Yangyang Liang for their constructive comments.

Additional Information and Declarations

Competing Interests

Author Contributions

Animal Ethics

Field Study Permissions

Data Availability

The authors declare there are no competing interests.

Huaxing Zhou conceived and designed the experiments, analyzed the data, prepared figures and/or tables, authored or reviewed drafts of the paper, and approved the final draft.

Yuting Hu conceived and designed the experiments, performed the experiments, authored or reviewed drafts of the paper, and approved the final draft.

He Jiang conceived and designed the experiments, authored or reviewed drafts of the paper, and approved the final draft.

Guoqing Duan analyzed the data, authored or reviewed drafts of the paper, and approved the final draft.

Jun Ling performed the experiments, authored or reviewed drafts of the paper, and approved the final draft.

Tingshuang Pan, Xiaolei Chen performed the experiments, prepared figures and/or tables, and approved the final draft.

Huan Wang and Ye Zhang analyzed the data, prepared figures and/or tables, authored or reviewed drafts of the paper, and approved the final draft.

The following information was supplied relating to ethical approvals (i.e., approving body and any reference numbers):

Procedures involving animals and their care were approved by the Animal Care and Use Committee of Anhui Academy of Agricultural Sciences under approval number 201003076.

The following information was supplied relating to field study approvals (i.e., approving body and any reference numbers):

Field experiments were approved by Fisheries Bureau of Anhui (project number: FB/AH 2017-10).

The following information was supplied regarding data availability:

The mitochondrial COI and Cyt b genes are available at GenBank: MN097948–MN098127 and MN098128–MN098307. Additional data is available in the Supplemental Files.

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
