# Peer review of "Population genetics of swamp eel in the Yangtze River: comparative analyses between mitochondrial and microsatellite data provide novel insights"

_PeerJ, doi:10.7717/peerj.8415_

## Round 0.1 · original submission · Major Revisions

Thank you for your submission regarding the population genetics of the swamp eel. Careful consideration of both reviewers' comments will improve this manuscript. Additionally, I have attached an annotated copy of your manuscript with additional comments.

I have suggested extensive English edits in the attached. Although I could not write a paper in a 2nd language myself, it is imperative that the language be readable (especially when you consider that many of your readers will not have English as their first language). Doing so will mean the widest audience for your work, so I suggest that you make the extra effort to make it as clear as possible.

Additionally, there are a few science issues that I raise:
1. Please refrain from using the word "population" when you mean sampling site. Indeed, your data suggest that the three sites on the main river are a single genetic group.
2. I suggest you change the term "mainstream" to "main stem" because the former has a different meaning in English.
3. The reviewers commented on the brevity of your introduction. Please expand this and include more information about previous genetic work on this species. Also, it would be useful to include introductory paragraphs on the ecology of this species (see review comments) and on the use of nuclear vs. mitochondrial markers - this is the main message of your paper so it will need to be highlighted.
4. I feel that the Nm estimations are really not informative here, since there are derived from Fst (which is reported). I would suggest removing this. If you feel strongly about keeping it, you need to make a better case as to why it provides additional information here.
5. You may want to consider doing an isolation by distance analysis (along the river distance, not Euclidean).
6. The big issue here is the way you have interpreted your unusual results with respect to mt vs nu markers. I think you need to explain this much better and in the context of the literature (there is much out there!). You mention that the life history of this species may be important to the interpretation of these data, but the dots are not well connected - please try to make your thoughts about this more explicit.

I would be pleased to re-evaluate your manuscript after all of these issues are addressed.

·

Basic reporting

First of all, I salute authors for conducting this high level genetic investigation of Monopterus albus. It is not so common to see such investigations carried out by researchers of one laboratory. I could go on by saying that this work was interesting and novatoric but unfortunately the policy of the journal is that such criteria are not evaluated. Therefore, I tried to prepare my review based on objective criteria. Despite the fact that I wrote many comments and suggestions how to improve your manuscript I can say that you need to prepare your manuscript only for minor-major changes.

Basic Reporting
In general, article text is written quite good but English of the manuscript could and should be improved due to high requirements of the journal. Introduction, and especially Discussion, could be improved too because not always sufficient information provided (see General comments section). There are no noticeable problems concerning the structure of the article. However, I was not able to see submitted sequences in GenBank using presented accession numbers and there is room for improvement for some figures and tables and raw data. Due to the fact that submission should include all results relevant to the hypothesis original MJ haplotype network files, at least rdf, should be also presented.

Experimental design

Authors presented original primary research within aims and scope of the journal. Research question (How population genetic structure of Monopterus albus is affected by this fish biology, i.e. sex reversal?) is relevant and meaningful but could be defined better. I think that after suggested changes in the manuscript (see General comments section) there will be no problems concerning research question, identification of the knowledge gap and statements how research fills this gap. In general, this investigation performed to a high technical and ethical standard. However, not all methods and procedures are described with sufficient information. It is really necessary to clarify why mtDNA COI and Cyt b markers were chosen.

Validity of the findings

As mentioned before, original data, at least rdf file, of MJ haplotype network should be provided. This would help to remove all doubts whether truly 47 mutation steps were detected between black dot (mv?) and H-67 haplotype. Ideally, one homologous sequence of species related to M. albus should be used for the same MJ haplotype network. This would reveal how many mutation steps should be expected between different related species from Monopterus genus. In general, major changes are not required in conclusions (see General comments section).

Additional comments

Submission
There are no problems concerning Section, Subjects, Keywords, Manuscript type, DNA data deposition statement, field study permits. However, I could not access the deposited data in GenBank. Experiments were necessary and ethical. Dear authors, despite the fact that it is only formality make sure that not some of You but all of You approved final draft of the manuscript. “( Monopterus albus )” I advise to change to “(Monopterus albus)”.

Abstract and Introduction
From Abstract it is not clear what these mainstream populations are but given in mind that Abstract have words limitation and this is later explained in the Introduction I think that it is not necessary to make big changes in Abstract text. In the Introduction text authors should explain why the population genetic research of the wild eel become a hot topic due to the development of the eel farming. It is not clear what Li et al (2013) determined after their research. Similarly, it is not clear how Liang et al (2016) investigated these six populations. That was natural populations? Finally, in Line 62 it is really necessary to clarify what mtDNA markers were chosen and why.

Minor changes:
Line 42: “nutritional” I advise to change to “nutritional value”.
Line 45 “eel” I advise to change to “swamp eel”. Otherwise some readers could be confused because “eel farming” is commonly used term for Anguillidae species.
Lines 47-48 “microsatellites” I advise to change to “DNA microsatellites”. Abbreviation ISSR should be explained.
Line 51 “researches” I advise to change to “investigations” or “studies”. It is not clear enough why the results of these investigations are important for swamp eel breeding.
Line 55 “normal” I advise to change to “most” or give particular examples of “normal” fishes. “preciseness enough” advise to change to “correct”.
Line 56 “base” I advise to change to “based”. Line 62 also.
Line 63 “Here, the nuclear” I advise to change to “Here, the results of nuclear loci” because not all DNA microsatellites represent nuclear genes. “comparative analyzed” I advise to change to “compared and analyzed”.
Line 64 “insight” I advise to change to “insights”.

Materials & Methods
How exactly 180 swamp eels were collected? With net? Other method? Due to the fact that in Lines 73-74 is first mention of DT, FC, HN, WW, GC and WJ samples it is necessary to explain more about these samples. As I understood, for all mtDNA and DNA microsatellite markers were used same PCR conditions. Is that right? Please write how mtDNA samples were prepared for DNA sequencing, with what apparatus DNA sequencing and microsatellite allele length determination were carried out. Singletons were sequenced with both primers from the same pair or using one primer? It is necessary to clarify what primers were used for DNA sequencing of mtDNA genes. It is also important to write in Materials & Methods that sequences of mtDNA genes were combined to one sequence. Why six populations were combined to one group in Line 97? What is the reason for this calculation? In Line 98 it is not clear what is the meaning of genetic variation. Alleles, Ho, He? Actually, Ho and He should be mentioned in this section and not only in Results section. Due to the fact that I could not see many possible links between different haplotypes in MJ haplotype network I suspect that not only MJ option was used. If that is the case then in Lines 110-111 additional reference should be given. Finally, in this section should be presented information about deletions and insertions in mtDNA genes and clarified whether this data were used in Dnasp and Network calculations.

Minor changes:
Line 72 “eel” I advise to change to “swamp eel”.
Line 80 “designed primers”. Newly designed? By Your research group or other researchers? It is necessary to present primers DNA sequences or references.
Line 81 References should be written in chronological order.
Line 92 Provide reference or link.
Line 96 “perform” I advise to change to “performed” or “conducted”.
Line 99 “estimated to” I advise to change to “estimated in order to”.
Line 104 “reflected” I advise to change to “reflect”.
Line 105 “revealed” I advise to change to “show”.
Line 111-113 Structure program is created for DNA microsatellite data analysis. Therefore, it should be mentioned that in this case genetic structure was analyzed using only DNA microsatellite data.

Results
It is not correct to state that genetic diversity of M. albus is high based on finding that from 180 sequences 86 were distinct haplotypes. High compared to what? Due to the fact that this is Results section it is enough to write “relatively high genetic diversity” but in Discussion it is necessary to compare obtained result with the genetic diversity of other species. Similarly, statement about high genetic diversity using 10 DNA microsatellite loci should be corrected. In Line 123 it is written “MJ method”. However, this is not clear enough how Clades were determined from MJ haplotype network. Actually, it is obvious that this time main criteria for determination of Clades was mutation steps between haplotypes. This information probably should be written in Materials & Methods. It would be interesting to know whether same result is obtained using NJ method but I will not ask this additional analysis. Despite the fact that haplotype 5 (H-5) is most common in the current study that does not mean that this haplotype is the ancestral haplotype. This is only an assumption. It would be interesting to find out whether sequence of related species to M. albus would group with H-5 or other haplotypes in haplotype network. Once again I will not require this additional analysis during this revision. What I will require is the examples from literature of determined huge Fst values (~0.9431) in the context of intraspecific genetic diversity in Discussion. What alternatives to gene flow could be given in Lines 151-152?

Minor changes:
Lines 123-124 “contained five groups”. Please clarify. Five from six populations?
Line 128 In case this is about genetic differentiation, link to Fst results needed. If not then no changes required.
Line 130 “was range” I advise to change to “ranged”.
Line 131 “result” I advise to change to “results”.
Line 132 “population” I advise to change to “populations”.
Line 133 link to Fst and Nm results needed.
Line 134 “main stream” I advise to change to “mainstream”.
Lines 137-138 “significant negative” I advise to change to “significant (p<0.01) but negative”.
Line 139 “while” I advise to change to “because”.
Line 140 text should be corrected because one FC Tajima D value was significant.
Line 142 “estimated” I advise to change to “indicated”.
Line 149 “Structure results” link to results needed.
Line 150 “respected” I advise to change to “represented”, “indicated” or “shown”.
Line 156 “result” I advise to change to “obtained result”. “suggested” I advise to change to “suggested that”.

Discussion
It would be interesting to see possible explanations why Hd and Pi values of FC population are smallest. Similarly, explanations why Pi and Ho values of DT population are highest. In Lines 164-165 text should be modified because it is not correct to write it this way. Actually, during this study obtained Hd = 0.6620-0.9793 and Pi = 0.0017-0.0148 while Liang et al (2016) obtained Hd = 0.708 and Pi = 0.002. So, what explanation could be given for greater Pi values during this study compared to Liang et al (2016) research? It is necessary to clarify and expand text in Lines 171-172, 177-178, 197-200, 204, 215. Indeed, explain these terms: “sex-biased dispersal”, “genetic admixture”, “incomplete inheritance of ancestral polymorphisms”, “selective pressure regulated”. Also improve English in Lines 204, 215-216, 216-218. Finally, compare your results with results of additional genetic investigations of M. albus.

Minor changes:
Lines 166-167 “significant” I advise to change to “crucial” or “noticeable”.
Line 169 Need links to results.
Line 173 “expansion” Where?
Lines 175-176 “common fish” Give particular examples. In what environment movement is weak? In water or on ground?
Lines 176-177 “other species” Clarify from what taxa? Same order or higher taxa?
Lines 181-182 “population” I advise to change to “populations”. Species name should be indicated. Alternative: “tributary population structure” I advise to change to “genetic structure of swamp eel”.
Lines 190-191 Need to write obtained Fst value.
Lines 191-194 Need to write obtained Fst values.
Line 195 “studies”. In case it is only the current research, I advise to change to “study”. In case it is also other studies, provide references.
Line 200 “This study” I advise to change to “During this study”.
Line 201 “mainstream populations” I advise to change to “mainstream population”.
Line 205 “suggested” I advise to change to “suggest that”.
Lines 212-214 “was suffered” I advise to change to “could experience”. It is not clear how many eggs produce swamp eels and common fishes.

Conclusions
As mentioned earlier, it is not correct to just write “high genetic diversity”.

Minor changes:
Lines 220-221 “demographic genetic variation” Explain. Maybe it is reasonable to write “demographic and genetic variations”?
Line 223 “Each tributary population” Only investigated during this work or all?
Lines 224-226 Text after “population genetic structure” should be removed or appear in new sentence.
Lines 226-228 “insight” I advise to change to “insights”. “population genetic research” I advise to change to “population genetic structure and research”.

References
All references in text could be found in reference list.

Figures and Tables
Figure 2 This figure should be improved by adding information about smallest (n=1) and largest (n=?) circles.“methods” I advise to change to “method”. “number of species” I advise to change to “number of sequences”. Black dots represent mv?
Figure 3 I advise to not write COI and Cyt b because sequences of these genes were combined. It is not indicated what species history.
Table 1 could be combined with Figure 1 but I will not require that. In case it is not enough support information that these six samples truly are six populations, “populations” I advise to change to “samples”.
Table 2 Add total values. It would be better to write “mitochondrial data” or “mitochondrial sequence” and not “mitochondrial genes” because there were no calculations of genetic diversity quantitative parameters of separate mtDNA genes.

Minor changes:
Figure 4 “gentic” I advise to change to “genetic”.
Table 4 “diaonal” I advise to change to “diagonal”. It is necessary to write what populations.
Table 5 “test” I advise to change to “tests”.

Raw Data
I advise to add information which individual COI and Cyt b sequences represent which haplotypes.

·

Basic reporting

The manuscript is generally well-written and presents interesting and relevant data.
The introduction could, in my view, be very much improved. The authors cite the relevant literature (in what concerns previous phylogenetic studies) but fail to provide important information on the biology of the species and its habitat. In my opinion, life-history (e.g. dispersal of larvae and adults) and environmental parameters (e.g. dynamics of the Yangtze River) are of crucial importance to understand the results. Some details are introduced in the discussion but in my view that should be done in a straightforward line of reasoning since the introduction.
Methods are adequate and results are sound but in my view the discussion also needs improvement mainly in the interpretation of the results and relevant literature.

Experimental design

My main concern relates with the use of the population concept. In this study populations are equivalent to sampling locations and not related with the phylogeographic consistency. Perhaps that should be revaluated throughout the manuscript, including the statement in lines 168-171.
Additionally, also in my opinion the authors should, if possible and for comparative purposes, retrieve the sequences already made available in Genbank for this sampling locations. That would also improve the data set and provide a wider comparative approach at least for mitochondrial markers. Considering mitochondrial markers with concatenate them since the beginning? We cannot therefore assess the diversity for each marker…

Concerning the conclusions they should be toned down because the results, in my view, do not support them entirely.

Validity of the findings

no comment

Additional comments

The manuscript “Genetic research of six swamp eel (Monopterus albus) populations in the Yangtze River based on mitochondrial and microsatellite markers”, written by Zhou and colleagues, focuses on the phylogeographic structure of this highly commercially species on the referred river.

The swamp eel is a very commercial species which has already been phylogenetically evaluated with various markers. The present study provides a comparative approach between different markers using 180 individuals from six eel populations in the Yangtze River, namely two mitochondrial genes and ten microsatellite loci.

Main results reveal:

- high diversity for all populations;
- significant differentiation for three tributary populations (probably due to seasonal cut-off;
- strong gene flow among the mainstream populations (no isolation by distance);
- discrepancy between mitochondrial and microsatellite results.

The manuscript is generally well-written and presents interesting and relevant data.
The introduction could, in my view, be very much improved. The authors cite the relevant literature (in what concerns previous phylogenetic studies) but fail to provide important information on the biology of the species and its habitat. In my opinion, life-history (e.g. dispersal of larvae and adults) and environmental parameters (e.g. dynamics of the Yangtze River) are of crucial importance to understand the results. Some details are introduced in the discussion but in my view that should be done in a straightforward line of reasoning since the introduction.
Methods are adequate and results are sound but in my view the discussion also needs improvement mainly in the interpretation of the results and relevant literature.

My main concern relates with the use of the population concept. In this study populations are equivalent to sampling locations and not related with the phylogeographic consistency. Perhaps that should be revaluated throughout the manuscript, including the statement in lines 168-171.
Additionally, also in my opinion the authors should, if possible and for comparative purposes, retrieve the sequences already made available in Genbank for this sampling locations. That would also improve the data set and provide a wider comparative approach at least for mitochondrial markers. Considering mitochondrial markers with concatenate them since the beginning? We cannot therefore assess the diversity for each marker…

Concerning the conclusions they should be toned down because the results, in my view, do not support them entirely.

Minor comments

Line 52-53: “However, as the typical sex reversal fish, Monopterus albus has its unique life history (Liu, 1944; Qu, 2018).” Perhaps it should be “However, as a typical sex reversal fish, Monopterus albus has a unique life history (Liu, 1944; Qu, 2018).
Line 72: replace 180 by One hundred and eighty at the beginning of the sentence.

I sincerely hope that my suggestions can help the authors to improve the paper.

Joana Robalo
(jrobalo@ispa.pt)

---

## Round 0.2 · Minor Revisions

Thank you for your careful attention to reviewer comments. I do think the manuscript is much improved.

Please address R1's new comments and make appropriate amendments.

Additionally, I have attached a PDF asking for minor clarifications/amendments. Finally, please address the following:

Major issues
On lines 276-283 You have used Brito (2007)’s correction to compare nuclear and mitochondrial estimates of Fst. First, you state that this accounts for differences in mutation rates but, in reality, this correction accounts for differences in effective population sizes between the two marker classes. Second, Brito (2007) states that this formula assumes an equal sex ratio, which is not the case with your study species. Third, when I calculate this, I get a corrected mtDNA Fst of 0.134, not 0.345. Most importantly, I do not believe this formula is appropriate for use in a protogynous species. If you do want to use such a correction, please find a formula that has been adjusted for protogynous species, or provide a novel development of an appropriate correction (e.g. with simulations). If you do not wish to do this, I suggest you remove this from your manuscript.

You have not discussed dispersal strategies in this manuscript. It would be useful to include what is known about dispersal because if females spawn, become male and then disperse, the differences in population structure between the markers would be explained. Please address this possibility in the manuscript.

On lines 164-165, you suggest that BSP indicates a main stem population decline. I think you have misinterpreted this result. The x-axis on the BSP is MYA, so that the most recent time is closest to the y-axis. This indicates that there has been a recent expansion, not decline. Please amend.

Minor issues
Figures 2 and 3 – please spell out abbreviations (e.g. “MJ”, “mv”, “HPD”)

·

Basic reporting

Last time I wrote many comments and suggestions for authors. Consequently, this time I will try to be more concise. I really like that it was helpful and how authors solved revision problems. This is was good learning experience for authors and me. The overall quality of the manuscript increased significantly.

Basic Reporting
In general, I had no problems with English of the manuscript but English language is not my native language. Therefore, I advise Editor to evaluate whether current English of the manuscript is sufficient for journal requirements. I checked presented MJ haplotype network files and found that in general there is no problem with this data. However, I noticed one problem regarding interpretation of this data: haplotype network configuration doesn't support idea that haplotype H-47 belong to Clade C. Actually, H-47 is only two mutations away from H-30 that belong to different Clade but 24 mutational steps away from H-42. This problem should be fixed before acceptance.

Experimental design

Authors improved manuscript quality and now I think that there are no problems concerning research question, identification of the knowledge gap and statements how research fills this gap. All methods and procedures are described with sufficient information with the exception of information about calculations regarding deletions and insertions. However, I want to ask why authors decided to reduce number of DNA microsatellite loci from 10 to 8.

Validity of the findings

In general, after revision conclusions part looks better.

Additional comments

Congratulations. Great work. You really improved your manuscript quality. I really like how you solved revision problems. Some questions remained unanswered but I have no problem with that because it is not important from the perspective of the current manuscript. I consider that your work should be published in this journal after minor but essential changes.

Main Issue
Haplotype network configuration doesn't support idea that haplotype H-47 belong to Clade C. Actually, H-47 is only 2 mutations away from H-30 that belong to Clade D but 24 mutational steps away from H-42 (Clade C). This problem should be fixed before acceptance.

Abstract, Introduction and M&M
I wrote that it was not necessary to make big changes in the Abstract text but I have to admit that after revision it looks better. I also see significant good improvement in the Introduction. It is good that now in the Introduction is explained that two genes sequences were combined to one but I think that the same thing should be done in M&M where is first mention about these genes. You can argue that it is explained later in M&M where mentioned DNASTAR but in order to avoid confusion for the readers I advise to make these changes. You convinced me that you used only MJ option. However, this leads to my suggestion to not write "parsimony network" because MP option was not used. I still have to ask to present information in the manuscript regarding deletions and insertions during different calculations. What is the reason behind reduction of available data from 10 to 8 DNA microsatellites?

Results, Discussion, Conclusions
As I understood the non-significant values of SSD of main stem and DT indicate that sudden expansion could not be rejected. Therefore, significant values of SSD of FC and HN indicate that sudden expansion could be rejected. Is that right? If yes then what explanation you can give it for FC and HN SSD results? This 0.46 MYA was calculated by different way than data presented in Figure 3? Thank you for answers regarding huge Fst values but I still have to ask the examples from literature (not necessarily of Monopterus in investigated basin but animal species in general, preferably fish species) of determined huge Fst values (~0.9431) in the context of intraspecific genetic diversity in Discussion. It is true that mtDNA is maternally inherited but there are rare cases where animals mtDNA could be inherited paternally. I advise you to try to investigate this in your future Monopterus studies.

Figures and Tables
As I understood in Figure 2 lines without numbers between different haplotypes mean that it is only one mutational step between them. Is that correct? That should be explained to reader. In Figure 3 it is not explained what HPD (readers should understand every picture and table without munuscript text). Explain to readers what SSD mean in Table 4. Also change "mainstream" to "main stem".

Minor changes:
Line 33 "cause" I advise to change to "caused".
Line 50 "over exploitation" I advise to change to "overexploitation".
Line 98 "m mol/LM" it seems that it should be: "m mol/L".
Line 102 "Genotype" I advise to change to "Genotypes".
Line 248 "factors" I advise to change to "factor".

·

Basic reporting

The manuscript "Population genetics of swamp eel in the Yangtze River: comparative analyses between mitochondrial and microsatellite data provide novel insights" is now in its second review. In my opinion the manuscript is very much improved. Nevertheless, it still presents some english imprecisions (e.g check the conclusions).
It should have a final revision by a native speacker.
Concerning my previous reviews, all points were rebutted accordingly.
Therefore my opinion is that, after english corrections, the manuscript may be published in its present form.

Experimental design

no comment

Validity of the findings

no comment

Additional comments

The manuscript "Population genetics of swamp eel in the Yangtze River: comparative analyses between mitochondrial and microsatellite data provide novel insights" is now in its second review. In my opinion the manuscript is very much improved. Nevertheless, it still presents some english imprecisions (e.g check the conclusions).
It should have a final revision by a native speacker.
Concerning my previous reviews, all points were rebutted accordingly.
Therefore my opinion is that, after english corrections, the manuscript may be published in its present form.

---

## Round 0.3 · accepted · Accept

Thanks very much for your diligence in editing this manuscript. It is greatly improved and will advance our understanding of the population genetics of this species and, more widely, of protogynous species.